## [Reviewer comments · Royal Society Open Science]

Review History

RSOS-172413.R0 (Original submission)

Review form: Reviewer 1

Is the manuscript scientifically sound in its present form?

Yes

Are the interpretations and conclusions justified by the results?

No

Is the language acceptable?

Yes

Is it clear how to access all supporting data?

Yes

Do you have any ethical concerns with this paper?

No

Have you any concerns about statistical analyses in this paper?

Yes

Recommendation?

Major revision is needed (please make suggestions in comments)

Comments to the Author(s)

The authors use a double digest RAD protocol to analyse the population structure of *Dascyllus trimaculatus* in the Indian Ocean. Overall, the lab protocols are well conducted, and the population structure methods are fairly standard and appear well conducted. The main problem I see, is with the narrative of the paper, that is slightly incoherent with the general purpose of the study. It's not clear what the authors are trying to achieve with their focus on neutral versus outlier in this particular case. The analyses using all loci show a very similar pattern to the outlier, apart from the outliers separating NRS from DJI in the DAPC analyses (but not in the structure analyses). The outlier analyses didn't provide much inference regarding potential adaptation, and as the authors acknowledge founder effects confound the issue. I feel like the authors would be best presenting the data from all SNPs first, and then present the neutral vs outlier comparison as a means to show that at least some of the separation signal appears to be driven by the outliers. This would fit better with a discussion where the potential for adaption is raised, but not possible to confirm from the present data. On this note, I think the authors could make a better effort in identifying potential genes in the vicinity of the outlier SNPs. The results on this are difficult to interpret as presented, and there is not detail on how these were identified.

I also agree with previous reviewers disagreeing with conclusions that this study shows the power of RADseq. Apart from this being well established by now, the number of SNPs is overall very low, particularly those that were identified as outliers. Power is thus likely to be low for a RADseq study, so authors really shouldn't be trying to argue for the power in their dataset. I also feel that there is very little detail on the analyses at times. Some information on methods and results are clearly lacking to allow a reader to assert whether the authors statements are accurate or not. I've included some more information on my specific comments below.

Therefore, my assessment is that the manuscript has potential to be published in Royal Society Open Science, but not in its current form. The authors need to address some of the weaknesses mentioned, but crucially need to improve the narrative of the paper to make it more coherent and focused on the questions the authors can address with this study (patterns of geographic structure in the Indian Ocean). I agree that this is an interesting region for marine biogeography which has receive relatively little attention, however I don't feel the manuscript emphasises this strongly enough in its present form. I also agree with the authors comment that, even though the study does not address a specific conservation concern, it can be used as a driver for better conservation policies locally. There are several cases known where research papers were the trigger for better conservation efforts, and this could be one of those cases. Therefore, I'm recommending rejection encouraging resubmission, and I encourage the authors to address the limitations of their manuscript more thoroughly and provide a strong resubmission.

Specific comments:

Lines 68-71: I feel this is overstated, given that differentiation is only found in the outlier loci

Lines 113-115: the Evanno et al transformation was designed for a very specific case when Ln values constantly increase with higher levels of K. Therefore, using this method as a standard is

not the correct way to assess the correct value of K from a structure run. Having said that, from the results it seems that the Evanno et al modification was not used, at least for the neutral dataset, as with the Evanno et al method, $K=1$ cannot ever be the most supported level of K.

Lines 127-128: there is no information whether Bonferroni correction was applied. From Table 1 I can see that the alpha was set to the standard 0.05, which suggests no Bonferroni correction (although how this is implemented varies depending on software). However, pairwise F_{st} like this always requires Bonferroni correction, so the authors need to make sure that Bonferroni correction was applied.

Lines 132-133: this sentence is self-contradictory, and does not match the results shown. DAPC clearly shows Oman separates from all others, but the structure plot does not show this distinction. Therefore, DAPC and structure results are not in agreement.

Lines 133-134: The authors need to show the slope of the IBD regression lines. Otherwise, not possible to assess the strength of IBD.

Lines 194-196: There is no information on how this was determined, but this is needed to understand exactly what was done and what these results mean.

Lines 211-312: I agree with a previous reviewer that his sentence is overstated. RADseq are widely known to provide more resolution on a number of analyses. I don't think the new phrasing really addresses the comment. It still sounds like the authors are suggesting their study shows the usefulness of RADseq, when this is now generally agreed on.

Review form: Reviewer 2 (James Hogan)

Is the manuscript scientifically sound in its present form?

Yes

Are the interpretations and conclusions justified by the results?

Yes

Is the language acceptable?

Yes

Is it clear how to access all supporting data?

Not Applicable

Do you have any ethical concerns with this paper?

No

Have you any concerns about statistical analyses in this paper?

No

Recommendation?

Accept with minor revision (please list in comments)

Comments to the Author(s)

Review of Salas et al. - Using outlier loci to elucidate patterns of gene flow in a coral reef fish
 Author: J. Derek Hogan, Associate Professor, Texas A&M University – Corpus Christi

This paper uses RAD sequencing to study the population genetics of a common coral reef fish in the Indian Ocean and looks to test hypotheses about the biogeography of the region. This is a fine paper that ultimately supports previous findings of biogeography in the Indian Ocean. The paper is generally well written and there are no major flaws in the methods, and the results are appropriately interpreted given the data and analyses. There is no reason not to publish this study.

My specific comments to improve the paper are below:

Title – the novelty of RAD and outlier loci to study patterns of genetic structure in non-model organisms is not great. I suggest refocusing the title to reflect the results of the paper rather than focus on methods which have become common place.

Introduction:

1. L 34 – 48 - This background is too generic and a bit out of date as a justification for studies of reef fish genetics now. Maybe 5 years ago this would have been novel enough. I think a much more compelling justification for this study is a detailed study of biogeography and genetic discontinuity in the Indian Ocean. There is enough work done now to devise a solid hypothesis for how genetic patterns in *Dascyllus* should manifest themselves across the Indian Ocean basin. This paper should be steeped in that literature and attempt to carve out a niche and forward a new understanding (or support for an existing hypothesis) about genetic biogeography in the IO.
2. L 54 – 59 - You should use this literature to devise a hypothesis and then determine the novelty of your study (first to use RAD?) then test the biogeographic hypothesis with your data and see if you support or refute the existing paradigm.
3. L 61 – 62 - Anything unique about *Dascyllus* compared to other species that have been studied in the IO so far that you might expect different results than previous studies?
4. L 62 – 63 - Is that typical of other species that have been studied in the IO? How does this species compare to others that have been studied so far? Why did you choose this particular fish?
5. L 63 – 64 - So dispersal may be restricted like other anemone specialists (i.e., that pesky nemo).
6. L 68 – 69 - What leads you to that hypothesis if all other fishes are showing panmixia? PLD shorter? Life-history/habitat specialization? Whats different about *Dascyllus* that leads you to believe it will be different?
7. L 70 – 71 - Is this the first study to use SNPs to look at genetic biogeography of the IO?

Discussion:

1. L 194 – 196 - This doesnt seem as plausible as drift effects. IBD is inherently a neutral pattern. Also the patterns of environmental variables in this case would not likely produce an IBD pattern via non-neutral processes. North vs. South yes, but not IBD (at least not from what I can tell looking as Fig S3).
2. L 198 - By isolation you mean Drift, correct?

Conclusions:

1. This conclusion section is redundant with the Discussion. Unless it is required by the journal I suggest removing it altogether.
2. L 211 – 213 - I think this is not a novel statement any longer. Basically every RAD paper could say this, but its getting old fast.

Review form: Reviewer 3

Is the manuscript scientifically sound in its present form?

No

Are the interpretations and conclusions justified by the results?

No

Is the language acceptable?

Yes

Is it clear how to access all supporting data?

Yes

Do you have any ethical concerns with this paper?

No

Have you any concerns about statistical analyses in this paper?

Yes

Recommendation?

Major revision is needed (please make suggestions in comments)

Comments to the Author(s)

In their descriptive manuscript, Salas et al investigate population genetic structure in *Dascyllus trimaculatus* in the Western Indian Ocean and the Red Sea. They use RAD data and compare “neutral” and “outlier” loci, and detect subtle but significant genetic structure between the NWIO and the WIO provinces.

While the study is purely descriptive and mostly relevant to the species being studied, most of the analyses are solid and well established and hence I think the study would be worthy of publication in Royal Society Open Science. However I do have some concerns over the outlier analysis performed.

FDIST is known to be prone to false positives, particularly when the real demographic history deviates from the island model (see for example Whitlock and Lotterhos, *American Naturalist* 2015, Vol, 186). Furthermore, using a p-value of 0.01 one would expect > 10 outliers just by chance. In that respect, a non-over conservative correction for multiple tests would be in order (for example, just using the r function “p.adjust” the authors could calculate false discovery rate?). I have a problem with analyzing separately a bunch of loci which could simply represent the right tail of a neutral distribution of F_{st} . Perhaps the authors should include more robust tests, such as OutFLANK and/or the FLK test developed by Bohnomme et al (*Genetics*. 2010 Sep; 186(1): 241–262.). Each of these method may be sensitive to different biases, but loci identified by different methods may be good candidate outliers. I think this would be required before the manuscript can be published.

Perhaps the most could be gained using the unsupervised test developed by Kempainen (*Molecular Ecology Resources* (2015) 15, 1031–1045) based on Linkage disequilibrium network analysis (LDna). This would allow to identify loci in high LD that could be associated with both phylogeographic structure and local adaptation. Once clusters of loci in high LD are identified, PCA can be carried out with loci in each cluster separately. Many of these clusters will reflect biogeography (i.e. they will separate a single population or a group of populations based on geographic structure) but some may group individuals in different ways. These could be candidate for local adaptation (although of course the authors should not overreach their conclusions). While I do not consider this analysis as a *conditio sine qua non* for publication, I strongly encourage the authors to have a go at it. If the authors find interesting groups of loci with patterns contrasting with bio-geographic patterns, then they could potentially look for association with environmental variables.

In addition, I find some statements perhaps superfluous. It is very well known (and not just in model organisms) that increasing number of loci can help in finding subtle patterns of population structure that are characteristic of large marine populations, and that in marine populations (given their large N_e and gene flow) selection is more efficient. Is that truly necessary to repeat? This sort of statement is ever more present in the introduction of many marine population

genetics paper. It can also be misleading in some way. While high N_e means that selection is more efficient, it also means that LD decays very fast around selected loci (unless they are in regions of low recombination, or in chromosomal rearrangements). Hence, low density SNPs are likely to miss the vast majority of targets of selection (see for example discussion in Hemmer-Hansen et al., *Biol. Bull.* 227: 117–132), unless they occur in regions of low recombination/chromosomal rearrangements.

Similarly, I think genetics is not a good proxy of population connectivity in demographic terms, even in age of NGS and whole genome sequencing, and particularly if using metrics such as F_{st} . Genetic connectivity is in itself very important for conservation and management, and so it's identifying local adaptation. But as the introduction is written it seems as if we get more loci we can get estimates that are meaningful in demographic terms, which is very often not the case. My suggestion would be to get rid of some of these "trite" statements and say something more about the biology/ecology of the species, as this is a descriptive paper mostly of interest to coral reef scientists and more specifically to the ones interested in this species. But again, this is just an advice that the authors may feel free to ignore.

The authors could also explore a bit more their data. For example, what are patterns of diversity across the range? Metrics like heterozygosity and nucleotide diversity (π) and Tajima's D could also help getting a better picture of the phylogeographic history of the species.

Decision letter (RSOS-172413.R0)

05-Dec-2018

Dear Ms Salas,

The editors assigned to your paper ("Using outlier loci to elucidate patterns of gene flow in a coral reef fish") have now received comments from reviewers. We would like you to revise your paper in accordance with the referee and Associate Editor suggestions which can be found below (not including confidential reports to the Editor). Please note this decision does not guarantee eventual acceptance.

Please submit a copy of your revised paper before 28-Dec-2018. Please note that the revision deadline will expire at 00.00am on this date. If we do not hear from you within this time then it will be assumed that the paper has been withdrawn. In exceptional circumstances, extensions may be possible if agreed with the Editorial Office in advance. We do not allow multiple rounds of revision so we urge you to make every effort to fully address all of the comments at this stage. If deemed necessary by the Editors, your manuscript will be sent back to one or more of the original reviewers for assessment. If the original reviewers are not available, we may invite new reviewers.

When submitting your revised manuscript, you must respond to the comments made by the referees and upload a file "Response to Referees" in "Section 6 - File Upload". Please use this to

document how you have responded to the comments, and the adjustments you have made. In order to expedite the processing of the revised manuscript, please be as specific as possible in your response.

- Data accessibility

If you wish to submit your supporting data or code to Dryad (<http://datadryad.org/>), or modify your current submission to dryad, please use the following link:
<http://datadryad.org/submit?journalID=RSOS&manu=RSOS-172413>

- Competing interests

- Authors' contributions

- Acknowledgements

- Funding statement

on behalf of Professor Michael Bruford (Associate Editor) and Kevin Padian (Subject Editor)
openscience@royalsociety.org

Comments to Author:

Reviewers' Comments to Author:

Reviewer: 1

Comments to the Author(s)

The authors use a double digest RAD protocol to analyse the population structure of *Dascyllus trimaculatus* in the Indian Ocean. Overall, the lab protocols are well conducted, and the population structure methods are fairly standard and appear well conducted. The main problem I see, is with the narrative of the paper, that is slightly incoherent with the general purpose of the study. It's not clear what the authors are trying to achieve with their focus on neutral versus outlier in this particular case. The analyses using all loci show a very similar pattern to the outlier, apart from the outliers separating NRS from DJI in the DAPC analyses (but not in the structure analyses). The outlier analyses didn't provide much inference regarding potential adaptation, and as the authors acknowledge founder effects confound the issue. I feel like the authors would be best presenting the data from all SNPs first, and then present the neutral vs outlier comparison as a means to show that at least some of the separation signal appears to be driven by the outliers. This would fit better with a discussion where the potential for adaption is raised, but not possible to confirm from the present data. On this note, I think the authors could make a better effort in identifying potential genes in the vicinity of the outlier SNPs. The results on this are difficult to interpret as presented, and there is not detail on how these were identified.

I also agree with previous reviewers disagreeing with conclusions that this study shows the power of RADseq. Apart from this being well established by now, the number of SNPs is overall very low, particularly those that were identified as outliers. Power is thus likely to be low for a RADseq study, so authors really shouldn't be trying to argue for the power in their dataset. I also feel that there is very little detail on the analyses at times. Some information on methods and results are clearly lacking to allow a reader to assert whether the authors statements are accurate or not. I've included some more information on my specific comments below.

Therefore, my assessment is that the manuscript has potential to be published in Royal Society Open Science, but not in its current form. The authors need to address some of the weaknesses mentioned, but crucially need to improve the narrative of the paper to make it more coherent and focused on the questions the authors can address with this study (patterns of geographic

structure in the Indian Ocean). I agree that this is an interesting region for marine biogeography which has received relatively little attention, however I don't feel the manuscript emphasises this strongly enough in its present form. I also agree with the authors' comment that, even though the study does not address a specific conservation concern, it can be used as a driver for better conservation policies locally. There are several cases known where research papers were the trigger for better conservation efforts, and this could be one of those cases. Therefore, I'm recommending rejection encouraging resubmission, and I encourage the authors to address the limitations of their manuscript more thoroughly and provide a strong resubmission.

Specific comments:

Lines 68-71: I feel this is overstated, given that differentiation is only found in the outlier loci

Lines 113-115: the Evanno et al transformation was designed for a very specific case when L_n values constantly increase with higher levels of K . Therefore, using this method as a standard is not the correct way to assess the correct value of K from a structure run. Having said that, from the results it seems that the Evanno et al modification was not used, at least for the neutral dataset, as with the Evanno et al method, $K=1$ cannot ever be the most supported level of K .

Lines 127-128: there is no information whether Bonferroni correction was applied. From Table 1 I can see that the alpha was set to the standard 0.05, which suggests no Bonferroni correction (although how this is implemented varies depending on software). However, pairwise F_{st} like this always requires Bonferroni correction, so the authors need to make sure that Bonferroni correction was applied.

Lines 132-133: this sentence is self-contradictory, and does not match the results shown. DAPC clearly shows Oman separates from all others, but the structure plot does not show this distinction. Therefore, DAPC and structure results are not in agreement.

Lines 133-134: The authors need to show the slope of the IBD regression lines. Otherwise, not possible to assess the strength of IBD.

Lines 194-196: There is no information on how this was determined, but this is needed to understand exactly what was done and what these results mean.

Lines 211-312: I agree with a previous reviewer that his sentence is overstated. RADseq are widely known to provide more resolution on a number of analyses. I don't think the new phrasing really addresses the comment. It still sounds like the authors are suggesting their study shows the usefulness of RADseq, when this is now generally agreed on.

Reviewer: 2

Comments to the Author(s)

Review of Salas et al. - Using outlier loci to elucidate patterns of gene flow in a coral reef fish

Author: J. Derek Hogan, Associate Professor, Texas A&M University - Corpus Christi

This paper uses RAD sequencing to study the population genetics of a common coral reef fish in the Indian Ocean and looks to test hypotheses about the biogeography of the region. This is a fine paper that ultimately supports previous findings of biogeography in the Indian Ocean. The paper is generally well written and there are no major flaws in the methods, and the results are appropriately interpreted given the data and analyses. There is no reason not to publish this study.

My specific comments to improve the paper are below:

Title – the novelty of RAD and outlier loci to study patterns of genetic structure in non-model organisms is not great. I suggest refocusing the title to reflect the results of the paper rather than focus on methods which have become common place.

Introduction:

1. L 34 – 48 - This background is too generic and a bit out of date as a justification for studies of reef fish genetics now. Maybe 5 years ago this would have been novel enough. I think a much more compelling justification for this study is a detailed study of biogeography and genetic discontinuity in the Indian Ocean. There is enough work done now to devise a solid hypothesis for how genetic patterns in *Dascyllus* should manifest themselves across the Indian Ocean basin. This paper should be steeped in that literature and attempt to carve out a niche and forward a new understanding (or support for an existing hypothesis) about genetic biogeography in the IO.
2. L 54 – 59 - You should use this literature to devise a hypothesis and then determine the novelty of your study (first to use RAD?) then test the biogeographic hypothesis with your data and see if you support or refute the existing paradigm.
3. L 61 – 62 - Anything unique about *Dascyllus* compared to other species that have been studied in the IO so far that you might expect different results than previous studies?
4. L 62 – 63 - Is that typical of other species that have been studied in the IO? How does this species compare to others that have been studied so far? Why did you choose this particular fish?
5. L 63 – 64 - So dispersal may be restricted like other anemone specialists (i.e., that pesky nemo).
6. L 68 – 69 - What leads you to that hypothesis if all other fishes are showing panmixia? PLD shorter? Life-history/habitat specialization? Whats different about *Dascyllus* that leads you to believe it will be different?
7. L 70 – 71 - Is this the first study to use SNPs to look at genetic biogeography of the IO?

Discussion:

1. L 194 – 196 - This doesnt seem as plausible as drift effects. IBD is inherently a neutral pattern. Also the patterns of environmental variables in this case would not likely produce an IBD pattern via non-neutral processes. North vs. South yes, but not IBD (at least not from what I can tell looking as Fig S3).
2. L 198 - By isolation you mean Drift, correct?

Conclusions:

1. This conclusion section is redundant with the Discussion. Unless it is required by the journal I suggest removing it altogether.
2. L 211 – 213 - I think this is not a novel statement any longer. Basically every RAD paper could say this, but its getting old fast.

Reviewer: 3

Comments to the Author(s)

In their descriptive manuscript, Salas et al investigate population genetic structure in *Dascyllus trimaculatus* in the Western Indian Ocean and the Red Sea. They use RAD data and compare “neutral” and “outlier” loci, and detect subtle but significant genetic structure between the NWIO and the WIO provinces.

While the study is purely descriptive and mostly relevant to the species being studied, most of the analyses are solid and well established and hence I think the study would be worthy of publication in Royal Society Open Science. However I do have some concerns over the outlier analysis performed.

FDIST is known to be prone to false positives, particularly when the real demographic history deviates from the island model (see for example Whitlock and Lotterhos, *American Naturalist* 2015, Vol, 186). Furthermore, using a p-value of 0.01 one would expect > 10 outliers just by chance. In that respect, a non-over conservative correction for multiple tests would be in order (for example, just using the r function “p.adjust” the authors could calculate false discovery

rate?). I have a problem with analyzing separately a bunch of loci which could simply represent the right tail of a neutral distribution of F_{st} . Perhaps the authors should include more robust tests, such as OutFLANK and/or the FLK test developed by Bohnomme et al (Genetics. 2010 Sep; 186(1): 241–262.). Each of these method may be sensitive to different biases, but loci identified by different methods may be good candidate outliers. I think this would be required before the manuscript can be published.

Perhaps the most could be gained using the unsupervised test developed by Kempainen (Molecular Ecology Resources (2015) 15, 1031–1045) based on Linkage disequilibrium network analysis (LDna). This would allow to identify loci in high LD that could be associated with both phylogeographic structure and local adaptation. Once clusters of loci in high LD are identified, PCA can be carried out with loci in each cluster separately. Many of these clusters will reflect biogeography (i.e. they will separate a single population or a group of populations based on geographic structure) but some may group individuals in different ways. These could be candidate for local adaptation (although of course the authors should not overreach their conclusions). While I do not consider this analysis as a *conditio sine qua non* for publication, I strongly encourage the authors to have a go at it. If the authors find interesting groups of loci with patterns contrasting with bio-geographic patterns, then they could potentially look for association with environmental variables.

In addition, I find some statements perhaps superfluous. It is very well known (and not just in model organisms) that increasing number of loci can help in finding subtle patterns of population structure that are characteristic of large marine populations, and that in marine populations (given their large N_e and gene flow) selection is more efficient. Is that truly necessary to repeat? This sort of statement is ever more present in the introduction of many marine population genetics paper. It can also be misleading in some way. While high N_e means that selection is more efficient, it also means that LD decays very fast around selected loci (unless they are in regions of low recombination, or in chromosomal rearrangements). Hence, low density SNPs are likely to miss the vast majority of targets of selection (see for example discussion in Hemmer-Hansen et al , : Biol. Bull. 227: 117–132), unless they occur in regions of low recombination/chromosomal rearrangements.

Similarly, I think genetics is not a good proxy of population connectivity in demographic terms, even in age of NGS and whole genome sequencing, and particularly if using metrics such as F_{st} . Genetic connectivity is in itself very important for conservation and management, and so it's identifying local adaptation. But as the introduction is written it seems as if we get more loci we can get estimates that are meaningful in demographic terms, which is very often not the case. My suggestion would be to get rid of some of these “trite” statements and say something more about the biology/ecology of the species, as this is a descriptive paper mostly of interest to coral reef scientists and more specifically to the ones interested in this species. But again, this is just an advice that the authors may feel free to ignore.

The authors could also explore a bit more their data. For example, what are patterns of diversity across the range ? Metrics like heterozygosity and nucleotide diversity (π) and Tajima's D could also help getting a better picture of the phylogeographic history of the species.

Author's Response to Decision Letter for (RSOS-172413.R0)

See Appendix A.

RSOS-172413.R1 (Revision)

Review form: Reviewer 1

Is the manuscript scientifically sound in its present form?

Yes

Are the interpretations and conclusions justified by the results?

Yes

Is the language acceptable?

Yes

Is it clear how to access all supporting data?

Yes

Do you have any ethical concerns with this paper?

No

Have you any concerns about statistical analyses in this paper?

No

Recommendation?

Accept with minor revision (please list in comments)

Comments to the Author(s)

I think the narrative is much improved and more cohesive. The paper now reads very well, and highlights the study strengths very clearly. I also think that the authors have addressed my comments satisfactorily, and I'm happy with the revisions made.

There are just a few minor comments I have to improve the manuscript. One is that the authors should revise the writing carefully for minor sentence construction problems. Note that the writing is generally very good and clear, but there are only some minor imperfections that can be easily ironed out and make the writing a bit neater.

Thanks for including the IBD numerical results. Looking at those, it seems there is no IBD really, so I suggest the authors phrase their results and discussion accordingly. I understand the p-value is significant for outliers, but the slope is very low and the model fit is also poor, so personally I don't see strong evidence for IBD in either case.

Regardless, I feel this is a strong paper in support of an interesting and well carried out study. I'm therefore happy to recommend this paper be accepted for publication.

Review form: Reviewer 3

Is the manuscript scientifically sound in its present form?

Yes

Are the interpretations and conclusions justified by the results?

Yes

Is the language acceptable?

Yes

Is it clear how to access all supporting data?

Yes

Do you have any ethical concerns with this paper?

No

Have you any concerns about statistical analyses in this paper?

No

Recommendation?

Accept with minor revision (please list in comments)

Comments to the Author(s)

I think the authors did a good job revising the manuscript. I am still curious as to why the authors decided to present the outliers/neutral snps analyses separately, as the tests for outliers they perform are prone to false discoveries, and only one test identified the loci analyzed. It would perhaps have made more sense to present in the main paper the analyses with all the snps, and in the supplementary materials the neutral/outlier datasets. In any case, the authors do not overstate the importance of the outlier analyses, so i think that is not big issue. I do have a few minor comments that the authors should consider before the paper is published:

Line 19: there does not seem to be any test specifically targeted at detecting gene flow. Low F_{st} can be due to ongoing gene flow, or also to recent ancestry and large N_e or secondary contact. In large marine pops it may not be always straightforward to discern between these, and it can't be done just by looking at F_{st} .

Line 40: "Entity", would "region" be better?

Line 112-113: if PLD is a poor predictor of dispersal, while should it explain genetic connectivity?

Line 203: "discern the adaptive value". slightly odd phrasing, also even if matches were found it may be difficult to do so without phenotypic and environmental data

Line 256: the results do not necessarily suggest that gene flow is variable across the genome, though this is possible. For example, recombination rate variation can provide a sufficient explanation for islands of divergence, with no necessity for restricted gene flow due to, for example, environmental selection or Dobzhansky-Muller incompatibilities. Butti shows this nicely in a review/perspective paper: where recombination is low, local N_e is lower, and lower diversity creates islands of differentiation due to drift and background selection.

Burri, R. (2017), Interpreting differentiation landscapes in the light of long-term linked selection. *Evolution Letters*, 1: 118-131. doi:10.1002/evl3.14

Line 270: What about Figure S2, does analyses with all loci show significant IBD? The relationship seems obvious but the authors do not report stats for the analyses with all loci

Decision letter (RSOS-172413.R1)

12-Apr-2019

Dear Ms Salas:

On behalf of the Editors, I am pleased to inform you that your Manuscript RSOS-172413.R1 entitled "RADseq analyses reveal concordant Indian Ocean biogeographic and phylogeographic boundaries in a reef fish, *Dascyllus trimaculatus*" has been accepted for publication in Royal Society Open Science subject to minor revision in accordance with the referee suggestions. Please find the referees' comments at the end of this email.

The reviewers and Subject Editor have recommended publication, but also suggest some minor revisions to your manuscript. Therefore, I invite you to respond to the comments and revise your manuscript.

- Ethics statement

- Data accessibility

If you wish to submit your supporting data or code to Dryad (<http://datadryad.org/>), or modify your current submission to dryad, please use the following link:
<http://datadryad.org/submit?journalID=RSOS&manu=RSOS-172413.R1>

- Competing interests

- Authors' contributions

- Acknowledgements

- Funding statement

Because the schedule for publication is very tight, it is a condition of publication that you submit the revised version of your manuscript before 21-Apr-2019. Please note that the revision deadline will expire at 00.00am on this date. If you do not think you will be able to meet this date please let me know immediately.

Supplementary files will be published alongside the paper on the journal website and posted on

the online figshare repository (<https://figshare.com>). The heading and legend provided for each supplementary file during the submission process will be used to create the figshare page, so please ensure these are accurate and informative so that your files can be found in searches. Files on figshare will be made available approximately one week before the accompanying article so that the supplementary material can be attributed a unique DOI.

on behalf of Professor Michael Bruford (Associate Editor) and Kevin Padian (Subject Editor)
openscience@royalsociety.org

Associate Editor Comments to Author (Professor Michael Bruford):

Associate Editor: 1

Comments to the Author:

Thanks for making such a thorough revision. We are happy to accept the paper subject to the few remaining minor revisions requested.

Reviewer comments to Author:

Reviewer: 3

Comments to the Author(s)

I think the authors did a good job revising the manuscript. I am still curious as to why the authors decided to present the outliers/neutral snps analyses separately, as the tests for outliers they perform are prone to false discoveries, and only one test identified the loci analyzed. It would perhaps have made more sense to present in the main paper the analyses with all the snps, and in the supplementary materials the neutral/outlier datasets. In any case, the authors do not overstate the importance of the outlier analyses, so i think that is not big issue. I do have a few minor comments that the authors should consider before the paper is published:

Line 19: there does not seem to be any test specifically targeted at detecting gene flow. Low F_{st} can be due to ongoing gene flow, or also to recent ancestry and large N_e or secondary contact. In large marine pops it may not be always straightforward to discern between these, and it can't be done just by looking at F_{st} .

Line 40: "Entity", would "region" be better?

Line 112-113: if PLD is a poor predictor of dispersal, while should it explain genetic connectivity?

Line 203: "discern the adaptive value". slightly odd phrasing, also even if matches were found it may be difficult to do so without phenotypic and environmental data

Line 256: the results do not necessarily suggest that gene flow is variable across the genome,

though this is possible. For example, recombination rate variation can provide a sufficient explanation for islands of divergence, with no necessity for restricted gene flow due to, for example, environmental selection or Dobzhansky-Muller incompatibilities. Butti shows this nicely in a review/perspective paper: where recombination is low, local N_e is lower, and lower diversity creates islands of differentiation due to drift and background selection.

Burri, R. (2017), Interpreting differentiation landscapes in the light of long-term linked selection. *Evolution Letters*, 1: 118-131. doi:10.1002/evl3.14

Line 270: What about Figure S2, does analyses with all loci show significant IBD? The relationship seems obvious but the authors do not report stats for the analyses with all loci

Reviewer: 1

Comments to the Author(s)

I think the narrative is much improved and more cohesive. The paper now reads very well, and highlights the study strengths very clearly. I also think that the authors have addressed my comments satisfactorily, and I'm happy with the revisions made.

There are just a few minor comments I have to improve the manuscript. One is that the authors should revise the writing carefully for minor sentence construction problems. Note that the writing is generally very good and clear, but there are only some minor imperfections that can be easily ironed out and make the writing a bit neater.

Thanks for including the IBD numerical results. Looking at those, it seems there is no IBD really, so I suggest the authors phrase their results and discussion accordingly. I understand the p-value is significant for outliers, but the slope is very low and the model fit is also poor, so personally I don't see strong evidence for IBD in either case.

Regardless, I feel this is a strong paper in support of an interesting and well carried out study. I'm therefore happy to recommend this paper be accepted for publication.

Author's Response to Decision Letter for (RSOS-172413.R1)

See Appendix B.

Decision letter (RSOS-172413.R2)

03-May-2019

Dear Ms Salas,

I am pleased to inform you that your manuscript entitled "RADseq analyses reveal concordant Indian Ocean biogeographic and phylogeographic boundaries in the reef fish *Dascyllus trimaculatus*" is now accepted for publication in Royal Society Open Science.

on behalf of Professor Michael Bruford (Associate Editor) and Kevin Padian (Subject Editor)
openscience@royalsociety.org

Follow Royal Society Publishing on Twitter: [@RSocPublishing](https://twitter.com/RSocPublishing)
Follow Royal Society Publishing on Facebook:
<https://www.facebook.com/RoyalSocietyPublishing.FanPage/>
Read Royal Society Publishing's blog: <https://blogs.royalsociety.org/publishing/>

Appendix A

Comment	Action
Reviewer: 1 The authors use a double digest RAD protocol to analyse the population structure of Dascyllus trimaculatus in the Indian Ocean. Overall, the lab protocols are well conducted, and the population structure methods are fairly standard and appear well conducted. The main problem I see, is with the narrative of the paper, that is slightly incoherent with the general purpose of the study. It's not clear what the authors are trying to achieve with their focus on neutral versus outlier in this particular case.	We now modified the narrative of the paper. This can be seen mainly on the modifications made in the introduction. Earlier, the introduction was focusing on the use of Rad Seq and analysis with outliers to get improved resolution on population genetic studies, using Dascyllus trimaculatus in the Indian Ocean as a case study. Now, we have modified the introduction to be more coherent with the general purpose of the study. The purpose is to explore the genetic structure of a fish in the Indian ocean. Therefore, the introduction now reviews the biogeography and genetic breaks in the Indian Ocean, why this region needs more studies and what would we expect with Dascyllus trimaculatus (and why we chose it). We are no longer making generalizations about RadSeq and outlier analysis. We simply use different sets of loci to explore the genetic structure of that fish, and discuss potential role of selection vs. drift. Please see the changes in the narrative in the introduction.
The analyses using all loci show a very similar pattern to the outlier, apart from the outliers separating NRS from DJI in the DAPC analyses (but not in the structure analyses). The outlier analyses didn't provide much inference regarding potential adaptation, and as the authors acknowledge founder effects confound the issue. I feel like the authors would be best presenting the data from all SNPs first, and then present the neutral vs outlier comparison as a means to show that at least some of the separation signal appears to be driven by the outliers. This would fit better with a discussion where the potential for adaption is raised, but not possible to confirm from the present data	Thanks for your recommendation which will help with clarity. We modified the manuscript to present first the results with all loci, then neutral and then outliers. We left the figures the same because the figure 1 represents the contrasting results very well. The figures of all loci are still in supplementary materials.
On this note, I think the authors could make a better effort in identifying potential genes in the vicinity of the outlier SNPs. The results on this are difficult to interpret as presented, and there is not	We looked for potential genes (presented in the supplementary table S3), but did not find genes that could clearly explain an adaptive function related to the divergence we observed. Most loci

detail on how these were identified.	didn't have significant matches. The methods on how potential genes were identified are explained on lines 178-182.
I also agree with previous reviewers disagreeing with conclusions that this study shows the power of RADseq. Apart from this being well established by now, the number of SNPs is overall very low, particularly those that were identified as outliers. Power is thus likely to be low for a RADseq study, so authors really shouldn't be trying to argue for the power in their dataset.	We now removed this statement from the discussion and conclusions.
I also feel that there is very little detail on the analyses at times. Some information on methods and results are clearly lacking to allow a reader to assert whether the authors statements are accurate or not. I've included some more information on my specific comments below. Therefore, my assessment is that the manuscript has potential to be published in Royal Society Open Science, but not in its current form. The authors need to address some of the weaknesses mentioned, but crucially need to improve the narrative of the paper to make it more coherent and focused on the questions the authors can address with this study (patterns of geographic structure in the Indian Ocean). I agree that this is an interesting region for marine biogeography which has receive relatively little attention, however I don't feel the manuscript emphasises this strongly enough in its present form. I also agree with the authors comment that, even though the study does not address a specific conservation concern, it can be used as a driver for better conservation policies locally. There are several cases known where research papers were the trigger for better conservation efforts, and this could be one of those cases. Therefore, I'm recommending rejection encouraging resubmission, and I encourage the authors to address the limitations of their manuscript more thoroughly and provide a strong resubmission.	Thank you for recommendations. We will address the issues using each one of your specific comments below.
Lines 68-71: I feel this is overstated, given that differentiation is only found in the outlier loci	We removed the sentence "We hypothesize that Indian Ocean D. trimaculatus are not a single panmictic population, and that the increased resolution provided by SNPs will reveal genetic differentiation across the region", and replaced with lines 100-105, with a more specific explanation of why these populations may not be panmictic, and without overstatements about SNPs and their resolution.

Lines 113-115: the Evanno et al transformation was designed for a very specific case when Ln values constantly increase with higher levels of K. Therefore, using this method as a standard is not the correct way to assess the correct value of K from a structure run. Having said that, from the results it seems that the Evanno et al modification was not used, at least for the neutral dataset, as with the Evanno et al method, K=1 cannot ever be the most supported level of K.	We now clarified this on the methods, in the lines 42-45 of supplementary materials and methods: “The most likely number of clusters (K) was determined by a combination of methods, first by looking at the assignment of the individuals. If most individuals are admixed, then we would expect no population structure and a K of 1. If there was more than one cluster, we applied the Evanno method [9] using STRUCTURE HARVESTER [10].
Lines 127-128: there is no information whether Bonferroni correction was applied. From Table 1 I can see that the alpha was set to the standard 0.05, which suggests no Bonferroni correction (although how this is implemented varies depending on software). However, pairwise Fst like this always requires Bonferroni correction, so the authors need to make sure that Bonferroni correction was applied.	Bonferroni corrections are applied now, see line 185, Table 1, and Table S1.
Lines 132-133: this sentence is self-contradictory, and does not match the results shown. DAPC clearly shows Oman separates from all others, but the structure plot does not show this distinction. Therefore, DAPC and structure results are not in agreement.	We have now removed “which is in agreement with the DAPC results” from the sentence. It now only says “ STRUCTURE analysis indicated K=1 (Fig 1). This now appears in lines 234-235
Lines 133-134: The authors need to show the slope of the IBD regression lines. Otherwise, not possible to assess the strength of IBD.	The full equation, including slope and intercept have been included to help interpretation of our data. This is now on the figure title captions (see Figure 1 and Figure S2 captions).
Lines 194-196: There is no information on how this was determined, but this is needed to understand exactly what was done and what these results mean.	To clarify what was done, we added more information on the methods and results. The methods on how potential genes were identified are explained on lines 178-182, and the results explained on lines 200-203, and table S1.
Lines 211-312: I agree with a previous reviewer that his sentence is overstated. RADseq are widely known to provide more resolution on a number of analyses. I don't think the new phrasing really addresses the comment. It still sounds like the authors are suggesting their study shows the usefulness of RADseq, when this is now generally agreed on.	We modified the focus of the paper and highlighted other interesting aspects more related with the biogeography of the region

Reviewer: 2 Author: J. Derek Hogan, Associate Professor, Texas A&M University – Corpus Christi This paper uses RAD sequencing to study the population genetics of a common coral reef fish in the Indian Ocean and looks to test hypotheses about the biogeography of the region. This is a fine paper that ultimately supports previous findings of biogeography in the Indian Ocean. The paper is generally well written and there are no major flaws in the methods, and the results are appropriately interpreted given the data and analyses. There is no reason not to publish this study. My specific comments to improve the paper are below: Title – the novelty of RAD and outlier loci to study patterns of genetic structure in non-model organisms is not great. I suggest refocusing the title to reflect the results of the paper rather than focus on methods which have become common place.	Thank you for your comments! We have now change the title to “RADseq analyses reveal concordant Indian Ocean biogeographic and phylogeographic boundaries in a reef fish, Dascyllus trimaculatus” In the title, we address the results of the paper, but we still mention the use of RadSeq, because it is the first paper using these methods to compare indian ocean populations that includes the african coast, mozambique channel, chagos, arabian sea and red sea altogether, and we consider worth highlighting that.
Introduction: 1. L 34 – 48 - This background is too generic and a bit out of date as a justification for studies of reef fish genetics now. Maybe 5 years ago this would have been novel enough. I think a much more compelling justification for this study is a detailed study of biogeography and genetic discontinuity in the Indian Ocean. There is enough work done now to devise a solid hypothesis for how genetic patterns in Dascyllus should manifest themselves across the Indian Ocean basin. This paper should be steeped in that literature and attempt to carve out a niche and forward a new understanding (or support for an existing hypothesis) about genetic biogeography in the IO.	We now removed that information. The introduction has been largely modified, and it now includes a detailed account on biogeographic and genetic discontinuity in the Indian Ocean, and explains what do we expect to see in a species such as Dascyllus trimaculatus
2. L 54 – 59 - You should use this literature to devise a hypothesis and then determine the novelty of your study (first to use RAD?) then test the biogeographic hypothesis with your data and see if you support or refute the existing paradigm.	Thank you for your suggestion, we have now used that literature to present our biogeographic hypothesis.
3. L 61 – 62 - Anything unique about Dascyllus compared to other species that have been studied	We don’t necessarily expect different results than previous studies, but we do highlight the unique

in the IO so far that you might expect different results than previous studies?	attributes of Dascyllus trimaculatus on lines 81-96, and 109-110
4. L 62 – 63 - Is that typical of other species that have been studied in the IO? How does this species compare to others that have been studied so far? Why did you choose this particular fish?	We now mention on lines 76-79 that these attributes represent a typical reef damselfish. Lines 61-73 present a review of other studies in the Indian Ocean that can be used to compare. In the lines 75-112 we explain why we chose this fish.
5. L 63 – 64 - So dispersal may be restricted like other anemone specialists (i.e., that pesky nemo).	Unlike that pesky nemo it generally settles on anemones, but not always (I've seen it on acropora heads for example, but is rare). Due to that and the larger PLD we expect moderate genetic structure but not as much as in clownfishes. We have now added on line 79 "generally settle on anemones", so our readers know that they are not absolute anemone specialists.
6. L 68 – 69 - What leads you to that hypothesis if all other fishes are showing panmixia? PLD shorter? Life-history/habitat specialization? What's different about Dascyllus that leads you to believe it will be different?	Please see our new background on the lines 75-119, that leads us to believe that these populations are not panmictic
7. L 70 – 71 - Is this the first study to use SNPs to look at genetic biogeography of the IO?	At that scale, including the WIO and NWIO, yes. Recent studies have used SNPs to look at clownfish within the Red Sea and Arabian Peninsula, but not outside.
Discussion: 1. L 194 – 196 - This doesn't seem as plausible as drift effects. IBD is inherently a neutral pattern. Also the patterns of environmental variables in this case would not likely produce an IBD pattern via non-neutral processes. North vs. South yes, but not IBD (at least not from what I can tell looking at Fig S3).	This is right, we have added this to the discussion, lines 296-297. However IBD can sometimes be an artifact. So we still do not exclude the possibility of selection.
2. L 198 - By isolation you mean Drift, correct?	Yes, we have changed that to drift (now on line 306).
Conclusions: 1. This conclusion section is redundant with the Discussion. Unless it is required by the journal I suggest removing it altogether.	We removed the section, we left the first sentences only.
2. L 211 – 213 - I think this is not a novel statement any longer. Basically every RAD paper could say this, but it's getting old fast.	We have now removed that statement.

Reviewer: 3 In their descriptive manuscript, Salas et al investigate population genetic structure in Dascyllus trimaculatus in the Western Indian Ocean and the Red Sea. They use RAD data and compare “neutral” and “outlier” loci, and detect subtle but significant genetic structure between the NWIO and the WIO provinces. While the study is purely descriptive and mostly relevant to the species being studied, most of the analyses are solid and well established and hence I think the study would be worthy of publication in Royal Society Open Science. However I do have some concerns over the outlier analysis performed. FDIST is known to be prone to false positives, particularly when the real demographic history deviates from the island model (see for example Whitlock and Lottheros, American Naturalist 2015, Vol, 186). Furthermore, using a p-value of 0.01 one would expect > 10 outliers just by chance. In that respect, a non-over conservative correction for multiple tests would be in order (for example, just using the r function “p.adjust” the authors could calculate false discovery rate?). I have a problem with analyzing separately a bunch of loci which could simply represent the right tail of a neutral distribution of Fst. Perhaps the authors should include more robust tests, such as OutFLANK and/or the FLK test developed by Bohnomme et al (Genetics. 2010 Sep; 186(1): 241–262.). Each of these method may be sensitive to different biases, but loci identified by different methods may be good candicate outliers. I think this would be required before the manuscript can be published. Perhaps the most could be gained using the unsupervised test developed by Kempainen (Molecular Ecology Resources (2015) 15, 1031–1045) based on Linkage disequilibrium network analysis (LDna). This would allow to identify loci in high LD that could be associated with both phylogeographic structure and local adaptation. Once clusters of loci in high LD are identified, PCA can be carried out with loci in each cluster separately. Many of these clusters will reflect biogeography (i.e. they will separate a single population or a group of populations based on geographic structure) but some may group individuals in different ways. These could be	We are aware of the issues described, and this is why our paper is not primarily about adaptation, rather it describes genetic structure, and discuss the potential role of adaptation, among others. We suspect that these outliers reflect the end tail of the neutral distribution (However, we cannot discard that these have adaptive function) Following your suggestion, we now also applied FDR to the arlequin analysis, using p.adjust (thanks for the idea!). We now present analysis with three methods, and applying false discovery rates in all of them (see lines 163-175, 194-200). However, the analysis of genetic structure found the same patterns with 25 loci than with less loci after correcting for multiple tests. Since our results did not change the general patterns, we decided to use the set of loci obtained in arlequin without the FDR (see lines 205-208) The additional analyses suggested by reviewer no. 3, although very interesting, are time consuming and would not add much new information to the story under its current limitations (few loci outliers, no information about the adaptive functions, confounding factor that environmental variables closely follow biogeographic barriers). So we decided not to pursue these analysis in this paper... but we are eager to try with other projects and we are really thankful for these useful suggestions!!!

candidate for local adaptation (although of course the authors should not overreach their conclusions). While I do not consider this analysis as a conditio sine qua non for publication, I strongly encourage the authors to have a go at it. If the authors find interesting groups of loci with patterns contrasting with bio-geographic patterns, then they could potentially look for association with environmental variables.	
In addition, I find some statements perhaps superfluous. It is very well known (and not just in model organisms) that increasing number of loci can help in finding subtle patterns of population structure that are characteristic of large marine populations (given their large N_e and gene flow) selection is more efficient. Is that truly necessary to repeat? This sort of statement is ever more present in the introduction of many marine population genetics paper. It can also be misleading in some way. While high N_e means that selection is more efficient, it also means that LD decays very fast around selected loci (unless they are in regions of low recombination, or in chromosomal rearrangements). Hence, low density SNPs are likely to miss the vast majority of targets of selection (see for example discussion in Hemmer-Hansen et al , : Biol. Bull. 227: 117–132), unless they occur in regions of low recombination/chromosomal rearrangements.	We have removed these arguments from the introduction.
Similarly, I think genetics is not a good proxy of population connectivity in demographic terms, even in age of NGS and whole genome sequencing, and particularly if using metrics such as F_{st}. Genetic connectivity is in itself very important for conservation and management, and so it's identifying local adaptation. But as the introduction is written it seems as if we get more loci we can get estimates that are meaningful in demographic terms, which is very often not the case. My suggestion would be to get rid of some of these "trite" statements and say something more about the biology/ecology of the species, as this is a descriptive paper mostly of interest to coral reef scientists and more specifically to the ones interested in this species. But again, this is just an advice that the authors may feel free to ignore.	Thank you, these are great observations, and we are now making an effort to not overstate our conclusions towards demographic terms, but still highlighting the importance of identifying population boundaries. The introduction has been largely modified to account for that.

The authors could also explore a bit more their data. For example, what are patterns of diversity across the range ? Metrics like heterozygosity and nucleotide diversity (π) and Tajima's D could also help getting a better picture of the phylogeographic history of the species	These analysis of diversity and heterozygosity did not provide important information so we decided not to put them on the paper. We would like to calculate Tajima's D, but I could not find how to analyze with allele frequencies (only complete sequences). Please let us know if you know how to analyze with SNP allele frequency data.

Appendix B

Comment	Action
Reviewer: 3 I think the authors did a good job revising the manuscript. I am still curious as to why the authors decided to present the outliers/neutral snps analyses separately, as the tests for outliers they perform are prone to false discoveries, and only one test identified the loci analyzed. It would perhaps have made more sense to present in the main paper the analyses with all the snps, and in the supplementary materials the neutral/outlier datasets. In any case, the authors do not overstate the importance of the outlier analyses, so i think that is not big issue.	The three datasets tell a similar story, but the outliers show it more clearly than any other dataset. We like how the neutral and outlier datasets present the story visually. Even though outliers are most likely not under selection, we cannot completely refute that hypothesis and want to highlight those results. It may inspire research questions in this dynamic biogeographic region.
I do have a few minor comments that the authors should consider before the paper is published: Line 19: there does not seem to be any test specifically targeted at detecting gene flow. Low Fst can be due to ongoing gene flow, or also to recent ancestry and large Ne or secondary contact. In large marine pops it may not be always straightforward to discern between these, and it can't be done just by looking at Fst.	We agree that Fst patterns can be due to reasons other than gene flow (or lack thereof). The discussion goes over a variety of reasons for the genetic structure patterns we found, covering gene flow, demographic history, etc. In that line 19, we removed gene flow, it originally read: “Neutral loci revealed a signature of gene flow between distant populations, with weak genetic differentiation between...” Now it says: “Neutral loci revealed a signature of weak genetic differentiation between...”
Line 40: "Entity", would "region" be better?	Better word, thanks, we replaced the word “entity” with the word “region”.
Line 112-113: if PLD is a poor predictor of dispersal, while should it explain genetic connectivity?	Thanks... Actually, our primary intention here was to illustrate with the PLD study that the environmental gradient found in the Red Sea could have an effect on D. trimaculatus life history, and this in turn could affect its genetic structure... due to many reasons. Among those, one reason may be restricted dispersal due to

	shorter PLD, but there could be many other reasons, like changes in fitness either in the adult or larval stages that could drive selection. Now, to answer your question: Most studies show a lack of relationship between PLD and dispersal (unless PLD is too short)... however this relationship still needs to be tested in more detail. Most studies to date compare many species with different PLDs, but do not carefully factor in the geographic and temporal variation of PLD within a species... so we believe that the idea that PLD doesn't affect dispersal cannot be completely refuted. We missed a little more explanation there. Anyways, we were trying to say a lot of things, so we will simplify our argument to focus on the primary point. We replaced this text: In the Red Sea, the pelagic larval duration (PLD) of D. trimaculatus decreases as sea surface temperatures and food availability increase from north to south [33]. Although PLD's are poor predictors of dispersal [34], this might affect genetic connectivity between the Red Sea and adjacent regions. With: In the Red Sea, the pelagic larval duration (PLD) of D. trimaculatus, D. aruanus and D. marginatus decrease as sea surface temperatures and food availability increase from north to south [33]. These results suggest that local environment variation affects the life history of D. trimaculatus, which in turn might influence its genetic structure.
Line 203: "discern the adaptive value". slightly odd phrasing, also even if matches were found it may be difficult to do so without phenotypic and environmental data	We removed the phrase "therefore we cannot discern the adaptive value"

Line 256: the results do not necessarily suggest that gene flow is variable across the genome, though this is possible. For example, recombination rate variation can provide a sufficient explanation for islands of divergence, with no necessity for restricted gene flow due to, for example, environmental selection or Dobzhansky-Muller incompatibilities. Butti shows this nicely in a review/perspective paper: where recombination is low, local N_e is lower, and lower diversity creates islands of differentiation due to drift and background selection. Burri, R. (2017), Interpreting differentiation landscapes in the light of long-term linked selection. Evolution Letters, 1: 118-131. doi:10.1002/evl3.14	Thank you for your insight. We are adding a sentence to account for that alternative explanation. These results suggest that gene flow is variable across the genome, and may be affected by different processes and/or operate at different scales. However, its important to acknowledge that variation in recombination rates could also explain patterns of divergence across the genome [50].
Line 270: What about Figure S2, does analyses with all loci show significant IBD? The relationship seems obvious but the authors do not report stats for the analyses with all loci	Yes, it is significant, the stats are in the figure caption... We will make it more clear in the main results text. We also included the equation, because even though it is significant, there is not a strong model fit. “Figure S2. Isolation by distance. Mantel test $p=0.0310$, $r^2=0.3006$, $y = 2E-06x - 0.0007$.”
Reviewer: 1 Comments to the Author(s) I think the narrative is much improved and more cohesive. The paper now reads very well, and highlights the study strengths very clearly. I Iso think that the authors have addressed my comments satisfactorily, and I'm happy with the revisions made. There are just a few minor comments I have to improve the manuscript. One is that the authors should revise the writing carefully for minor sentence construction problems. Note that the writing is generally very good and clear, bu there are only some minor imperfections that can be easily ironed out and make the writing a bit	Thanks, we now fixed issues with sentence construction.

neater.	
Thanks for including the IBD numerical results. Looking at those, it seems there is no IBD really, so I suggest the authors phrase their results and discussion accordingly. I understand the p-value is significant for outliers, but the slope is very low and the the model fit is also poor, so personally I don't see strong evidence for IBD in either case. Regardless, I feel this is a strong paper in support of an interesting and well carried out study. I'm therefore happy to recommend this paper be accepted for publication.	We changed results interpretations accordingly. Lines 225-226 now read: Finally, there was statistically significant isolation by distance (IBD, Fig S2), but the slope is low and there isn't a strong model fit And lines 244-246: Outliers revealed significant isolation by distance (Fig 1), but again the slope is low and there isn't a strong model fit. We also reflect that change in the discussion.